# Machine Learning Prediction of Mean Arterial Pressure from the Photoplethysmography Waveform During Hemorrhagic Shock and Fluid Resuscitation

**DOI:** 10.3390/s25165035

**Published:** 2025-08-13

**Authors:** Jose M. Gonzalez, Saul J. Vega, Shakayla V. Mosely, Stefany V. Pascua, Tina M. Rodgers, Eric J. Snider

**Affiliations:** U.S. Army Institute of Surgical Research, JBSA Fort Sam Houston, San Antonio, TX 78234, USA; jose.m.gonzalez355.civ@health.mil (J.M.G.);

**Keywords:** deep learning, feature extraction, hemorrhage, machine learning, resuscitation

## Abstract

**Highlights:**

**What are the main findings?**
Different combinations of feature extraction methodologies and artificial intelligence models successfully predicted mean arterial pressure throughout a clinically relevant hemorrhage and resuscitation animal study.Manually extracted features fed into a long short-term memory network predicted mean arterial pressure with the highest accuracy in this study.

**What is the implication of the main finding?**
Manual feature extraction using clinically relevant waveform features, combined with deep learning architectures, can accurately predict mean arterial pressure using non-invasive sensors.This technology, with further maturation, has the potential to significantly improve pre-hospital trauma management, improving triage decisions and guiding hemorrhage and resuscitation.

**Abstract:**

We aimed to evaluate the non-invasive photoplethysmography waveform as a means to predict mean arterial pressure using artificial intelligence models. This was performed using datasets captured in large animal hemorrhage and resuscitation studies. An initial deep learning model trained using a subset of large animal data and was then evaluated for real-time blood pressure prediction. With the successful proof-of-concept experiment, we further tested different feature extraction approaches as well as different machine learning and deep learning methodologies to examine how various combinations of these methods can improve the accuracy of mean arterial pressure predictions from a non-invasive photoplethysmography sensor. Different combinations of feature extraction and artificial intelligence models successfully predicted mean arterial pressure throughout the study. Overall, manual feature extraction fed into a long short-term memory network tracked the mean arterial pressure through hemorrhage and resuscitation with the highest accuracy.

## 1. Introduction

Patient monitoring through standard vital signs is an essential aspect of clinical care [1]. In trauma and emergency situations, both in civilian and military contexts, measurements such as heart rate, temperature, respiratory rate and others can inform a first-responder or clinician on the evolving patient status to guide triage, resource allocation, and treatment decisions [2,3]. The vital sign of interest, blood pressure, is a fundamental vital sign used to assess overall cardiovascular function and perfusion of organs and tissues.

While a traditional cuff-based sphygmomanometer can be used to obtain sporadic blood pressure measurements, a critical limitation of this type of device is that it becomes ineffective and highly inaccurate in hypotensive patients, such as those presenting hemorrhagic injuries [4,5]. On the other hand, catheterization of a large artery is considered the “gold standard” method for continuous and accurate blood pressure monitoring [6]. This procedure is routinely performed in properly resourced emergency and intensive care units. However, in the pre-hospital setting, catheterizing an artery carries serious risks, including infection, thrombosis, accidental catheter dislodgement and additional bleeding [7,8,9]. Therefore, alternative, non-invasive approaches to measuring blood pressure can provide benefits to trauma casualties being cared for by clinicians. One such alternative that has been explored to estimate arterial blood pressure is through the use of photoplethysmography (PPG). A PPG sensor attached to the skin uses electro-optical techniques to measure the light absorption changes that occur due to the blood vessels thickening and relaxing with every heartbeat [10]. The low cost, small footprint and non-invasive nature of modern PPG transducers, paired with advances in machine learning (ML), has led to the development of algorithms for predicting blood pressure in the form of mean arterial pressure (MAP) from PPG waveforms.

Several studies have leveraged ML methodologies to estimate blood pressure from PPG waveforms, a trend cataloged in a recent systematic review by Pal et al. [11]. The authors evaluated a variety of PPG-based blood pressure estimation models across various complexities of artificial intelligence (AI). Concerns are presented about the homogeneity of the training, testing, and validation datasets as many manuscripts in the blood pressure estimation field of study commonly use data from the Multiparameter Intelligent Monitoring in Intensive Care (MIMIC) intensive care unit family of datasets. The lack of diverse datasets demonstrates concerns about generalizability in limited available datasets involving hypotension and trauma. Despite this concern, there have been lots of AI architecture innovations for predicting blood pressure. Chu et al. proposed a transformer-based architecture trained on data from the MIMIC-III database [12] to estimate systolic and diastolic pressures from PPG signals leveraging empirical mode decomposition and personalization. They were able to achieve accuracies satisfying clinical standards, but the systolic and diastolic pressures were predominately in ranges in normotensive and hypertensive ranges [13]. Similarly, Slapničar et al. successfully used a spectro-temporal ResNet and personalization to predict systolic and diastolic blood pressures from the PPG and the associated first and second derivative from a curated MIMIC-III subset [14]. Athaya and Choi made use of a U-Net based model architecture to reconstruct an arterial waveform from PPG using the datasets extracted from MIMIC [15] and MIMIC-III. This architecture used the arterial waveform prediction to assess systolic, diastolic, and MAP and was accurate per British Hypertension Society [16], but was once again limited to those blood pressure ranges [17]. Additionally, El and Kyriacou used PPG derived features fed into a long short-term memory (LSTM) and gated recurrent unit model to predict blood pressure. Features were derived from the MIMIC-II database [18] and they demonstrated that modeling temporal variation in selected morphological PPG features with the selected model architecture yielded performance that satisfies Advancement of Medical Instrumentation accuracy requirements for non-invasive blood pressure estimation [19]. These studies, while they illustrate various methodologies to predicting blood pressure from PPG, have largely been trained and validated on datasets that underrepresent hypotensive states. As such, there is an unmet need to assess PPG-based MAP prediction in trauma-relevant datasets, particularly during hemorrhagic shock.

In this research effort, we aimed to assess the real-time ability of an early iteration of a convolutional neural network (CNN) paired with a LSTM network to predict MAP in a proof-of-concept large animal hemorrhagic shock and resuscitation study. This was performed to provide evidence that non-invasive PPG sensors and ML can provide MAP predictions in real-time that trend with the gold standard MAP obtained with an arterial line. This focus on real-time MAP predictions during hemorrhagic states distinguishes this work from prior studies that primarily predict blood pressure in normo- or hypertensive conditions.

Secondarily, this study evaluated the effects of the addition of more training data to the original CNN-LSTM network, different feature extraction approaches, and different ML and deep learning (DL) models for the prediction of MAP from PPG waveforms. Specifically, three unique model architecture approaches were evaluated using high-fidelity physiological data from hemorrhagic shock and fluid resuscitation large animal studies. The first approach used the CNN-LSTM model architecture to automatically extract features from the PPG signal which the LSTM subsequently used to generate MAP predictions. The remaining two ML models consisted of a gradient-boosted decision tree (XGBoost), and a simple LSTM network. Unlike the hybrid CNN-LSTM model, these two did not use an automated feature extractor, but instead received features manually engineered from the PPG waveform samples. All approaches were compared to assess the effects of different architecture and feature extraction techniques on their accuracy to predict MAP on datasets obtained from trauma-relevant hemorrhage and resuscitation animal studies.

## 2. Materials and Methods

### 2.1. Animal Study

Research was conducted in compliance with the Animal Welfare Act, the implemented Animal Welfare regulations and the principles of the Guide for the Care and Use of Laboratory Animals [20]. The Institutional Animal Care and Use Committee at the United States Army Institute of Surgical Research approved all research. The facility where these research projects were conducted is fully accredited by the AAALAC international. The two studies [21,22] were designed to demonstrate the usage of various physiological closed-loop controllers for hemorrhagic shock resuscitation in swine (*Sus scrofa domestica*). Swine were chosen due to their physiological similarities to humans, specifically the cardiovascular system [23,24]. Each study utilized *n* = 12 four-month-old, intact, female Yorkshire crossbred swine with an average weight of 40 kg.

All the animals were maintained at a surgical plane of anesthesia and given Buprenorphine SR in each study. An arterial line was inserted into a carotid artery to monitor arterial blood pressure, and another arterial line was inserted into a femoral artery for all blood sampling. Total intravenous anesthesia was administered through a triple-lumen central venous catheter that was inserted into the jugular vein. Lastly, a venous line was inserted into a femoral vein for both hemorrhaging and resuscitation. A PPG sensor was placed into the mouth after the cheeks were shaved or onto the ear to capture distinct PPG waveforms. After completing the instrumentation of the subject, a laparotomy and surgical splenectomy was performed.

Next, animals were allowed to stabilize after the surgical event and baseline readings were captured. A controlled hemorrhage to a MAP of 35 mmHg was reached either directly (Study #1) or in a stepwise fashion (i.e., 75, 65, 55, 45, and 35 mmHg; Study #2). After reaching the 35 mmHg target, the swine were held in a hypovolemic state until a lactate value of ≥4 mmol/L was achieved, or 90 min had lapsed. After the hypovolemic hold was completed, the controller-based resuscitation was initiated. Whole blood was initially infused followed by lactated Ringer’s solution once reaching a target MAP (Study #1) or after one liter of whole blood (Study #2) was used. The controllers were configured to infuse fluid until MAP had reached either the pre-hemorrhage baseline pressure or 65 mmHg, whichever was lower, and then set to maintain this target MAP during a 60-min hold period. The experiments continued after the whole blood resuscitation, but that is outside of the scope of this study. Upon completion of the experiment, the swine were euthanized.

### 2.2. Real-Time Data Processing and MAP Prediction

We initially evaluated an ML model architecture in real-time during the animal study to determine the feasibility of non-invasive prediction of blood pressure for this application. Initially, we developed a software pipeline that could simulate the effects of real-time data streaming. This allowed a continuous PPG signal to be processed by the AI models and estimate MAP in real time. Pre-recorded PPG data sampled at 100 Hz paired with corresponding timestamps, was “played back” and fed to a mockup AI model. This was developed in Python 3.9 using multiprocessing pipes to process the data at an interval of 10 ms, allowing it to stream at the same rate that the signal was originally recorded. This allowed for debugging of the streamed MAP prediction process prior to real-time to the live animal study implementation.

Proof-of-concept of real-time MAP predictions was tested in a single swine subject during the described animal procedure (Study #2). A Dräger Infinity Delta XL patient monitor (Drägerwerk AG & Co. KGaA, Lübeck, Germany) was used to collect the continuous PPG signal from the experimental subject. A custom Python script designed with the same multiprocessing scheme described above for debugging was then used to record the PPG signal on a standard personal computer by capturing the signal at a rate of 100 Hz using a LabJack U3-HV data acquisition unit (LabJack Corp, Lakewood, CO, USA) interfaced via a USB port. MAP prediction on the PPG data in real-time used a CNN-LSTM model as described in Methods Section 2.3.3. It was trained with 20-s-long PPG segments using data captured during the first animal study. The estimated MAP values were recorded and displayed on the computer screen simultaneously; results were compared to gold standard arterial line data to quantify the error and correlation between gold standard and model predictions.

### 2.3. Machine and Deep Learning Model Methodology

In addition to proof-of-concept evaluation of MAP predictions in real-time, we compared the performance of various AI model architectures for this application. While the CNN-LSTM model type was used for real-time testing, we also evaluated traditional ML and LSTM models to assess their performance. These were trained using data from the first and second animal study. This section first provides an overview of ML and DL methods, followed by specifics on each of the model architectures used in this study.

#### 2.3.1. Overview of Machine Learning and Deep Learning

Classical ML and DL are subsets of the AI field that focuses on the development of algorithms that can be trained to recognize patterns or classify data by learning from input information. Classical ML models use statistical approaches to identify patterns in datasets to make predictions based on similar inputs. DL models, on the other hand, are a further specialized subset of ML that draw inspiration of how a biological brain works for development of their architecture for learning and making predictions [25]. A distinct characteristic of DL models is that they can be trained to learn features from raw data, potentially bypassing the need for performing feature engineering manually.

Many decision-tree centric classical ML models exist that can be applied to make predictions of desired measurements based on data inputs. They often make decisions by creating random vectors from an input vector. These vectors reach a node and make a decision at said node, resulting in a “tree” growing from the node. This can be repeated until the desired parameters are reached [26]. For this effort, we used an XGBoost ML model which take insights on cache patterns, data compression and sharding to build a scalable boosting system to make predictions [27]. Previous efforts have used this model type to predict blood pressure in clinical settings using dialysis specific biomarkers [28], as well as using non-invasive measurements for its prediction [29].

In addition to the ML model types, DL model architectures built with CNN, LSTM, and fully connected layers were evaluated. CNNs are a type of feedforward neural network that extracts features from the input data with convolution structures [30], differing from traditional feature extraction methods that require manual feature extraction [31,32,33]. LSTMs are a type of recurrent neural network that is able to learn about and remember sequential data over periods of time and use the stored information to make predictions.

#### 2.3.2. Data Processing

An overview of the preprocessing steps, feature extraction pipelines, and modeling approaches are illustrated in Figure 1. Figure 1 summarizes the workflow from recording the raw PPG signal and ground truth MAP values from the animal study were processed, separated into both automated and manual feature extraction pathways, and subsequently used to train and evaluate the CNN-LSTM, XGBoost, and LSTM models.

##### Automated Feature Extraction (AutoFE)

Swine hemorrhage and resuscitation data from animal studies were used for the development of two CNN-LSTM models. Analog PPG data originally recorded at 500 Hz was downsampled to 100 Hz, and filtered using a 2nd order Butterworth bandpass filter with cutoff frequencies of 0.5 and 10 Hz, as these have been shown to be effective at removal of noise artifacts in PPG signals [34]. Outliers in the PPG signal were identified by calculating the interquartile range (IQR) and any values that fell outside of 1.5 times the IQR in the positive and negative direction were replaced by linear interpolation using neighboring PPG values. After filtering the PPG signal, its 1st through the 4th derivatives were calculated similar to previous work using PPG derivatives to predict blood pressure [35]. Prior to feeding the PPG signal and its derivatives into the DL model for training and testing, the data were segmented into intervals of 20 s, resulting in input datasets consisting of the PPG signal, its derivatives, and 2000 rows of data. The CNN layer then extracted features from the inputs, which were then fed to the LSTM layer for the prediction of MAP.

##### Manual Feature Extraction (MFE)

The same PPG signals recorded during the animal studies were processed to also extract manually engineered features, which were then fed into XGBoost mode and a simple neural network connected to an LSTM network. Analog PPG was downsampled to 100 Hz as previously described. For this portion of the study, data processing and feature extraction was performed using the pyPPG toolbox developed by Goda et al. for Python [36]. The inputs for the toolbox were filtered using a 4th order Chebyshev Type II filter with cutoff frequencies of 0.5 and 12 Hz. Fiducial points were then identified by the pyPPG toolbox in the PPG signal and its first three derivatives, which it then used to calculate 102 features. We proceeded to use these features to predict MAP. Specifically for the LSTM input, the features grouped into 10-heartbeat-long segments. This resulted in an input for the LSTM consisting of 102 features per beat for a total of 10 beats. This interval was selected based as it had optimal performance compared to 5 and 20 beats. For the XGBoost model, each of the 102 features were associated with a MAP value at each beat and were used to predict MAP at each beat; thus, no segmentation of the data was necessary.

#### 2.3.3. Model Development

All AI models were implemented using Python 3.9. Deep learning models were built with Keras 3, while gradient boosting models were implemented using DMLC’s XGBoost framework. For DL experiments, the Adam optimized was used with a batch size of 64 and a maximum of 100 training epochs, with early stopping applied to prevent overfitting. Additional model architecture details and XGBoost parameters are shown in Figure 2. All experiments were conducted on a computer system equipped with a 12th Gen Intel Core i7–12700H CPU, 64 GB RAM, and an NVIDIA GeForce RTX 3070 Ti mobile GPU with 8 GB of dedicated VRAM (Lenovo, Morrisville, NC, USA).

##### AutoFE CNN-LSTM Model

The CNN-LSTM DL model developed for the prediction of MAP used the input previously described in the AutoFE section. Due to the limited availability of data, the “leave one subject out” (LOSO) cross-validation technique was used to train, validate and test the model. One swine was held out for completely blind testing of the model while the remaining swine were split into a 90% training/10% validation split to develop the model. The initial version of this CNN-LSTM model, henceforth named RT-CNN-LSTM, was used in a proof-of-concept, real-time, live prediction experiment. This was developed with a total of 11 swine, with the 12th swine being the real-time test. Upon gathering more swine hemorrhage and resuscitation data, an improved CNN-LSTM model was developed with a total of 24 swine—the same swine that was used in the real-time test was held out for blind testing for comparison purposes.

##### MFE XGBoost Model

The XGBoost ML model developed for the prediction of MAP used the input previously described in the MFE section. The LOSO technique was used to validate the model on a completely blind test subject, the swine subject that was used in the real-time experiment. Due to the nature of the XGBoost model not having a validation schema similar to DL models, the validation swine of the previous DL models were included in the training of the XGBoost model.

##### MFE LSTM Model

The LSTM DL model developed for the prediction of MAP used the input mentioned in the MFE section. Like previous model development, the LOSO cross-validation technique was employed for the validation of the LSTM model. The real-time swine subject was held out for completely blind testing while the remaining swine were split into a 90% training/10% validation split. The same training and validation animals used to develop the updated CNN-LSTM model (i.e., 24 swine total) were used here again to eliminate subject variability having an impact on model performance comparisons if different training/validation subjects were used.

#### 2.3.4. Performance Metrics

Different metrics were used to compare the ground-truth MAP values recorded by a patient monitor with the MAP predictions generated by all the AI models, which consist of three unique model architecture designs. The Pearson’s correlation coefficient (r) was calculated to measure the linear correlation between the predicted MAP values and the actual values. The coefficient of determination was then found by squaring the Pearson correlation coefficient (R^2^) to statistically measure how well the predicted MAP approximates the actual values. Secondly, the root mean squared error (RMSE) determined how far the predicted MAP values fell from the actual values. An accuracy score was calculated to determine how well the predicted MAP values fell within the acceptable range of the ground-truth MAP (±10 mmHg), as determined by international standards for blood pressure measuring devices [37]. Lastly, performance error (PE) and absolute performance error (APE) were calculated based on the difference between ground truth MAP and predicted MAP, relative to predicted MAP for each model for the test swine subject.

PE and APE data were used to calculate intrasubject variability for predictions. In addition, median performance error (MDPE) and median absolute performance error (MDAPE) were calculated. Statistically significant differences in predictions were determined using a non-parametric Kruskal–Wallis test, post hoc Dunn’s test where n was based on replicate predictions by each AI model for the single test subject. Approximately 200 to 250 predictions were generated for CNN and LSTM models while XGBoost generated more than 2500 predictions. Significant differences between MDPE or MDAPE for each of four model configurations was defined as *p*-values less than 0.05 and are denoted when appropriate.

## 3. Results

Performance for four different model results are presented, starting with initial proof-of-concept evaluation in real-time using the RT-CNN-LSTM model. Next, the three MAP prediction models made using data from both animal studies are presented—AutoFE CNN-LSTM, MFE XGBoost, and MFE LSTM models. Results for each are shown as correlation plots between prediction and ground-truth MAP and predicted and ground-truth MAP plots with respect to time. Performance metrics for each model are summarized as R^2^, RMSE, and Accuracy values (Table 1).

For the RT-CNN-LSTM model, performance was low with an R^2^ of 0.470, RMSE of 9.70 mmHg, and accuracy of 62.4%. Correlation plot (Figure 3A) highlights the large regions of the data that were not tracking accurately which is further reflect from data plots versus time (Figure 3B). The predictions often fell outside the acceptable tolerance during the hemorrhage region but still tracked the resuscitation well. The updated AutoFE CNN-LSTM model with a larger training dataset had an improved R^2^ of 0.666, a reduced RMSE of 8.5 mmHg, and a higher accuracy of 83.1%. Improved correlation was evident especially during the hemorrhage phase, but predictions were less accurate during the resuscitation phase of the study (Figure 3C,D).

For the MFE XGBoost model, performance was measured at an R^2^ of 0.724, an RMSE of 6.9 mmHg, and an accuracy of 88.2%. Correlation fit was reduced compared to the AutoFE CNN-LSTM, as evident by poorer prediction tracking at larger hypovolemic states and during the resuscitation phase (Figure 4A,B). Lastly, the MFE LSTM model performance was strong, with an R^2^ of 0.866, an RMSE of 5.9 mmHg, and an accuracy of 90.6%. Stronger correlation was evident across different hypovolemic levels as well as during the resuscitation phase of the experiment (Figure 4C,D).

Lastly, PE and APE values were calculated to assess errors spread across all predictions (Figure 5) to provide an additional approach to compare each model’s performance. Overall, the two MFE models had MDPE values closest to zero at −1.45% and 2.21% for MFE XGBoost and MFE LSTM, respectively. MFE XGBoost was significantly different from the other three models. For MDAPE, the lowest errors were the same two models at 5.78% and 7.25% for MFE LSTM and MFE XGBoost, respectively. The highest error was with RT-CNN-LSTM at 15.09% and had a wider spread of values. These results highlight how the MFE models have the strongest performance for this application and will be considered in further research and refinement.

## 4. Discussion

Vital sign monitoring is essential for proper tracking of a casualty’s status during initial triage, medical evacuation, and administration of treatment. In the pre-hospital setting, vital sign tracking is limited to non-invasive sensing devices. As these non-invasive modalities are limited in diagnostic capabilities, ML models can potentially aim to expand their utility by developing predictive models for invasive measurements, such as continuous blood pressure estimation. Here, the ML models shown allow for non-invasive blood pressure measurement during baseline, hemorrhage, and resuscitation events using in vivo datasets.

An initial proof-of-concept study evaluated the setup and prediction capability for use in real-time during a large animal study. The setup diverted PPG waveforms to a data acquisition device and computer for prediction and was able to maintain predictions at high fidelity under these real-time conditions, at a rate of 3 predictions per minute. Compared to non-invasive blood pressure cuffs, which can only provide measurements at intervals of approximately every 60 s, this is an improvement over the standard sphygmomanometer. Further, we have previously shown in swine, as have other studies shown in humans, that traditional blood pressure cuffs have higher variability and performance loss at hypovolemic blood volumes [22]. This is a problem for using this technology to triage injury severity and guide hemorrhage resuscitation. However, the proof-of-concept RT-CNN-LSTM model had a low R^2^ and accuracy while having a high RMSE, likely due to the small training dataset used to develop the prediction model.

Thus, we evaluated if a larger training dataset and different predictive model approaches could improve performance. This was unable to be tested in real-time as the data was retrospectively analyzed, but model performance was improved. The LSTM models using features manually extracted from the waveform performed the best with an R^2^ of 0.866, and the second best was the XGBoost using the same features with an R^2^ of 0.724. These models did not only have a strong goodness of fit but also predicted accurately, with 90.6% and 88.2% of their prediction falling within ±10 mmHg of the ground truth value. These improvements may be due to manually extracted features being physiologically informed and based on prior studies [36,38,39]. Conversely, allowing the AI to extract features itself likely results in less medically relevant features. However, future work needs to identify which MFE features were most important to model predictions to understand the underlying physiology and simplify model input by removing unnecessary features. In addition, these models need to be tested in real-time but initial evidence suggests that these model types are more suitable for real-time MAP tracking during hemorrhage and resuscitation events.

However, there are limitations to this current study. The preprocessing for the PPG signal in the AutoFE versus MFE noticeably differed which may impact the study results. The AutoFE setup mimicked the real-time test while the MFE made use of a toolbox that had more advanced PPG preprocessing. This is especially true as the MFE approach made use of curated features that were thought to be relevant to this application as opposed to letting the AI try and identify its own waveform trends. There is potential for the AutoFE approaches to improve if similar, more advanced preprocessing is used. In addition, blind testing was performed on a single swine for comparison purposes, gathering additional data for more blind testing will be required to further validate these models. Lastly, only the RT-CNN-LSTM was evaluated in real-time and data degradation during the streaming process may impact performance. The other model types explored will be evaluated in future animal studies in real-time to determine if their performance is similar during real-time implementation.

Next steps will include implementing this technology for driving hemorrhage resuscitation therapies. This will lower the skill threshold for these therapies by not requiring the placement of an arterial catheter, allowing their use at earlier roles of medical care. Second, more data will be implemented to make the models more robust and generalized for a wider range of applications, potentially even beyond hemorrhagic shock. Lastly, the optimized models will be evaluated in further real-time implementations during future large animal studies. This will provide further confirmation of the success of these models to predict blood pressure, a critical next step for translating this technology into the pre-hospital and combat casualty care environment.

## 5. Conclusions

In summary, this study provides evidence that clinically informed MFE can be used in combination with DL architectures to accurately predict MAP using non-invasive sensors. This technology may improve trauma care in both pre-hospital and acute care environments through how triage decisions are made after further testing and clinical translation. These findings serve to demonstrate that hemorrhage and resuscitation diagnosis and treatment, which is guided by MAP measures using invasive sensors, can be improved through translation of standard vital signs that are typically obtained using invasive sensors to non-invasive technologies.

## Figures and Tables

**Figure 1 sensors-25-05035-f001:**
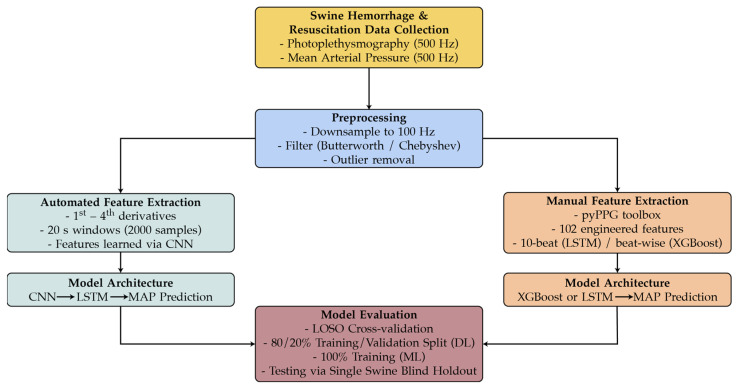
Data processing and modeling workflow for predicting MAP.

**Figure 2 sensors-25-05035-f002:**
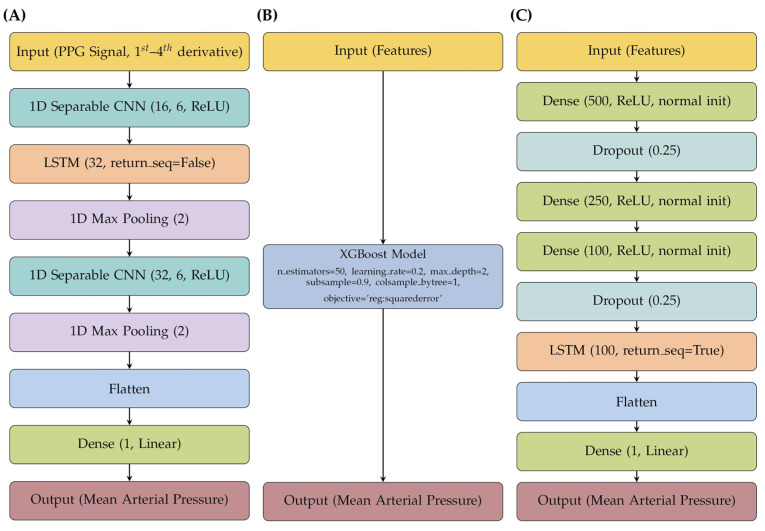
Comparison of model architectures for predicting MAP of (**A**) CNN-LSTM, (**B**) MFE XGBoost, and (**C**) MFE LSTM.

**Figure 3 sensors-25-05035-f003:**
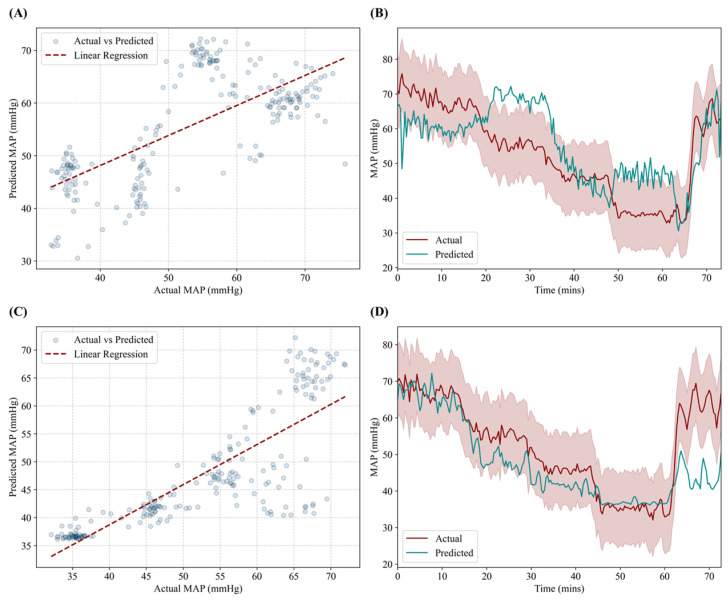
Performance results for the RT-CNN-LSTM and AutoFE CNN-LSTM models. (**A**,**C**) Correlation plot between predicted MAP and ground truth MAP for (**A**) RT-CNN-LSTM and (**C**) AutoFE CNN-LSTM models. Transparent points demonstrate congregation of points versus potential outliers. (**B**,**D**) Prediction and ground truth results plotted against time for the hemorrhage and resuscitation events for (**B**) RT-CNN-LSTM and (**D**) AutoFE CNN-LSTM models. Red shading represents allowable deviation from the true MAP value.

**Figure 4 sensors-25-05035-f004:**
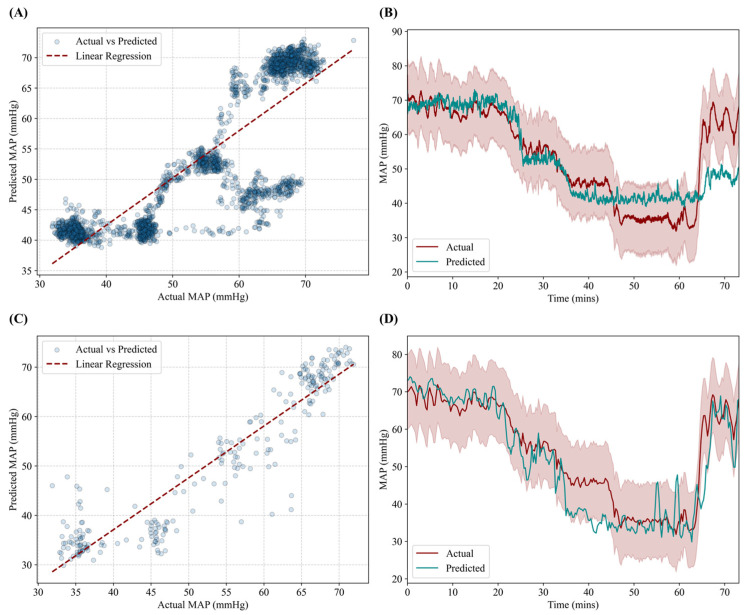
Performance results for the MFE XGBoost and MFE LSTM models. (**A**,**C**) Correlation plot between predicted MAP and ground truth MAP for (**A**) MFE XGBoost and (**C**) MFE LSTM models. Transparent points demonstrate congregation of points versus potential outliers. (**B**,**D**) Prediction and ground truth results plotted against time for the hemorrhage and resuscitation events for (**B**) MFE XGBoost and (**D**) MFE LSTM models. Red shading represents allowable deviation from the true MAP value.

**Figure 5 sensors-25-05035-f005:**
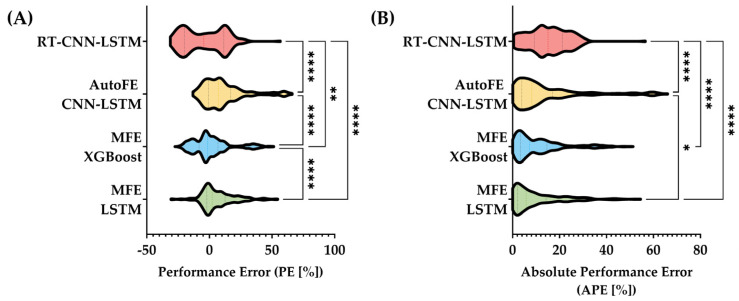
Summary of performance error and absolute performance error performance metrics for MAP prediction models. Distributions of (**A**) performance error and (**B**) absolute performance error are shown with median, first, and third quartile values for each model shown as lines on each dataset distribution. Statistical differences for each metric were determined by Kruskal–Wallis test, post hoc Dunn’s test. *p*-values less than 0.05 (*), 0.01 (**), and 0.0001 (****) are shown.

**Table 1 sensors-25-05035-t001:** Performance metrics comparison of models.

AI Model	R^2^	RMSE	Accuracy
RT-CNN-LSTM	0.470	9.70 mmHg	62.4%
CNN-LSTM	0.666	8.46 mmHg	83.1%
MFE XGBoost	0.724	6.94 mmHg	88.2%
MFE LSTM	0.866	5.87 mmHg	90.6%

## Data Availability

The data presented in this study are not publicly available because they have been collected and maintained in a government-controlled database located at the U.S. Army Institute of Surgical Research. This data can be made available through the development of a Cooperative Research and Development Agreement (CRADA) with the corresponding author.

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
