# Peer review of "Machine Learning Prediction of Mean Arterial Pressure from the Photoplethysmography Waveform During Hemorrhagic Shock and Fluid Resuscitation"

_sensors, 2025, doi:10.3390/s25165035_

Round 1
Reviewer 1 Report
Comments and Suggestions for Authors
The manuscript explores a clinically meaningful topic: non-invasive MAP prediction using PPG and machine learning in hemorrhagic shock and resuscitation scenarios. The strengths of the study include a systematic comparison of 4 models with different feature extraction methods (automated vs. manual) . There are several shortcomings in the study that need to be addressed.
1.Only the RT-CNN-LSTM model was evaluated in real-time. The other high-performance models (MFE LSTM and MFE XGBoost) need real-time testing to confirm their applicability in dynamic clinical environments .
2.The manuscript does not clarify which specific manual features contribute to the superior performance in line 214. Analyzing feature importance would improve the models' interpretability and clinical relevance.
Author Response
The manuscript explores a clinically meaningful topic: non-invasive MAP prediction using PPG and machine learning in hemorrhagic shock and resuscitation scenarios. The strengths of the study include a systematic comparison of 4 models with different feature extraction methods (automated vs. manual). There are several shortcomings in the study that need to be addressed.
We appreciate the reviewer taking the time to provide comments and suggestions for our manuscript. We hope our comments below address the issues, and that it is now suitable for publication.
1. Only the RT-CNN-LSTM model was evaluated in real-time. The other high-performance models (MFE LSTM and MFE XGBoost) need real-time testing to confirm their applicability in dynamic clinical environments.
Thank you for your comment, we agree. However, we do not have that data or resources to pursue that activity at this time. We used the real-time data to evaluate the others but full real time testing with more animals is not possible at this time but will be pursued in future work. We have detailed this as a limitation stating that the additional model architectures will need to be evaluated in future animal studies for real-time testing performance.
2. The manuscript does not clarify which specific manual features contribute to the superior performance in line 214. Analyzing feature importance would improve the models' interpretability and clinical relevance.
Currently, the AI models use all extracted features so this was not considered at this time but will be evaluated in future studies to identify what features are most relevant and important to AI model predictions. This has been more clearly mentioned in the discussion section now.
Reviewer 2 Report
Comments and Suggestions for Authors
The manuscript presents a relevant and timely study on predicting mean arterial pressure (MAP) using photoplethysmography (PPG) signals and machine learning (ML)/deep learning (DL) models in a swine model of hemorrhagic shock and resuscitation. The integration of ML /DL with non-invasive sensing is valuable for pre-hospital care. However, the manuscript requires major revisions before it can be considered for publication. Methodological clarity, fairness in model comparisons, stronger articulation of novelty, and improved writing quality are essential.
- The manuscript’s main novelty is not sufficiently emphasized. Please clarify whether it lies in Real-time application in hemorrhagic settings,Model architecture choices, Manual vs. automated feature extraction comparison, or Validation in trauma-relevant datasets. I recommend the authors expand the discussion in the introduction section comparing this study with prior work in PPG-based blood pressure estimation, particularly in hypotensive or trauma contexts. Additionally, I advise the authors justify the clinical significance of using a ±10 mmHg threshold as the accuracy benchmark, especially in high-risk hypotensive patients.
- In Section 2, four AI models were explained at high level, yet there is no figure illustration on the detailed model architectures and processing pipelines, which is critical for readers to interpret the AI model detail. I strongly recommend the authors add a graphical/schematic illustration figure on the presented AI model architecture and processing pipelines as the first figure to strengthen the presentation of the manuscript.
- In Section 2.2 and 2.3, AI model and data preprocessing were presented at high level. I advise the authors clearly describe the model architectures, including number of layers, hidden units, learning rates, optimizer choices, epochs, and batch sizes. In Section Section 2.3.2, having different prepossessing steps and filter designs for AutoFE and MFE models makes them hard to evaluate the AI model performance. I recommend the authors justify the rationale on implementing different preprocessing steps for these to models. Also, in Section 2.3, Please clarify how the 24 swine were divided into training, validation, and blind test groups across all models. Holding out a single swine for real-time blind testing may not be sufficient for robust validation. It is recommended to consider implementing k-fold or leave-one-out cross-validation across multiple subjects. Another missing part is the model training resource including the hardware, software resource used for the AI model training and the time cost. Please include those information as they are essential for anyone who wants to reproduce the work. Lastly, please provide rationale for choosing CNN-LSTM as the AutoFE model baseline, especially given its lower performance compared to MFE LSTM.
- In Section 3, the model performance metrics were evaluated at high level. I suggest the authors include statistical comparisons (e.g., confidence intervals, paired t-tests) between model performance metrics to support claims of superiority. Also, can you discuss why MFE models outperform AutoFE models — are manual features more physiologically meaningful? In the discussion, can you include more discussion on why AutoFE performs worse during resuscitation, and how feature relevance may shift across physiological states.
- For the reference and citations, I recommend the authors provide more critical comparisons with similar studies (e.g., references [11]–[14], [24]). Also, please ensure citations support the statements they are associated with, especially regarding previous failures of traditional cuffs in hypotensive cases. Additionally, please balance self-citations (e.g., [33], [34]) with independent literature for broader validation of approach and results.
- I cannot find any supplementary documents on the code related to this manuscript. I advise the authors include implementation or code details (e.g., GitHub link, pseudocode) to support reproducibility of model development.
Author Response
The manuscript presents a relevant and timely study on predicting mean arterial pressure (MAP) using photoplethysmography (PPG) signals and machine learning (ML)/deep learning (DL) models in a swine model of hemorrhagic shock and resuscitation. The integration of ML /DL with non-invasive sensing is valuable for pre-hospital care. However, the manuscript requires major revisions before it can be considered for publication. Methodological clarity, fairness in model comparisons, stronger articulation of novelty, and improved writing quality are essential.
We appreciate the reviewer taking the time to provide comments and suggestions for our manuscript. We hope our comments below address the issues, and that it is now suitable for publication.
The manuscript’s main novelty is not sufficiently emphasized. Please clarify whether it lies in Real-time application in hemorrhagic settings, Model architecture choices, Manual vs. automated feature extraction comparison, or Validation in trauma-relevant datasets. I recommend the authors expand the discussion in the introduction section comparing this study with prior work in PPG-based blood pressure estimation, particularly in hypotensive or trauma contexts. Additionally, I advise the authors justify the clinical significance of using a ±10 mmHg threshold as the accuracy benchmark, especially in high-risk hypotensive patients.
We are sorry the main novelty of the paper was not properly emphasized. It is much closer aligned to the “Real-time application in hemorrhagic settings” of the options the reviewer described, but the other aspects are touched on as well. We have clarified this primary and secondary intents for this study. We have added more comparisons to other studies in the introduction and more clearly justified the paper goal. The 10mmHg threshold is based on international standards for blood pressure measuring devices. This is referenced and mentioned in the methodology section 2.3.4.
In Section 2, four AI models were explained at high level, yet there is no figure illustration on the detailed model architectures and processing pipelines, which is critical for readers to interpret the AI model detail. I strongly recommend the authors add a graphical/schematic illustration figure on the presented AI model architecture and processing pipelines as the first figure to strengthen the presentation of the manuscript.
Thank you for your suggestion, we have added an additional figure to more clearly demonstrate the overall pipeline of the methodology used in this paper. In addition, we have added a figure depicting the model architectures for the three models presented in this study.
In Section 2.2 and 2.3, AI model and data preprocessing were presented at high level. I advise the authors clearly describe the model architectures, including number of layers, hidden units, learning rates, optimizer choices, epochs, and batch sizes. In Section Section 2.3.2, having different prepossessing steps and filter designs for AutoFE and MFE models makes them hard to evaluate the AI model performance. I recommend the authors justify the rationale on implementing different preprocessing steps for these to models. Also, in Section 2.3, Please clarify how the 24 swine were divided into training, validation, and blind test groups across all models. Holding out a single swine for real-time blind testing may not be sufficient for robust validation. It is recommended to consider implementing k-fold or leave-one-out cross-validation across multiple subjects. Another missing part is the model training resource including the hardware, software resource used for the AI model training and the time cost. Please include those information as they are essential for anyone who wants to reproduce the work. Lastly, please provide rationale for choosing CNN-LSTM as the AutoFE model baseline, especially given its lower performance compared to MFE LSTM.
Thank you for your suggestions, we have added a preamble in the model development section with the requested information as well as a figure for the different model architectures. The rationale on the different preprocessing steps was due to trying to match the real-time study, which is a n of 1, and doing more advancements from there. We have described it as a limitation of the study in the discussion section.
In Section 3, the model performance metrics were evaluated at high level. I suggest the authors include statistical comparisons (e.g., confidence intervals, paired t-tests) between model performance metrics to support claims of superiority. Also, can you discuss why MFE models outperform AutoFE models — are manual features more physiologically meaningful? In the discussion, can you include more discussion on why AutoFE performs worse during resuscitation, and how feature relevance may shift across physiological states.
Thanks for the suggestion. Unfortunately due, to the small sample size during real-time evaluation in the proof-of-concept study, a number of statistical approaches are not possible. We have added performance error and absolute performance error metrics to the results and have provided statistical analysis on an individual replicate data point level for the single real-time test dataset. These results have been added and described in the methods.
As for the MFE vs. AutoFE differences, yes, the MFE approaches are more physiological relevant as the features used were curated by the end-user with clinically motivated features. This has been expanded on in the discussion.
For the reference and citations, I recommend the authors provide more critical comparisons with similar studies (e.g., references [11]–[14], [24]). Also, please ensure citations support the statements they are associated with, especially regarding previous failures of traditional cuffs in hypotensive cases. Additionally, please balance self-citations (e.g., [33], [34]) with independent literature for broader validation of approach and results.
We have added a more detailed literature review in the introduction to better motivate the study and compare to prior work. We hope this addresses this point.
I cannot find any supplementary documents on the code related to this manuscript. I advise the authors include implementation or code details (e.g., GitHub link, pseudocode) to support reproducibility of model development.
There is no supplementary code due to limitations with government owned data per our US Army Institute of Surgical Research location. However, these data and code can be made available via agreement with our institute which is mentioned in the data sharing section.
Reviewer 3 Report
Comments and Suggestions for Authors
In this manuscript, the authors aim to non-invasively predict MAP during hemorrhagic shock and resuscitation using PPG in combination with ML models.
The primary concern with the paper is that the use of PPG to predict MAP via ML or DL has been extensively explored. The authors themselves cite 5 prior studies that used similar approaches. The main innovation claimed by the authors is the application of this prediction in a hemorrhagic hypotension setting. However, this distinction is relatively minor and not convincingly presented as substantially different from prior work. Specifically, to appropriately motivate the study, the authors should explicitly articulate what and why previous methods fail/would fail to achieve in this specific setting and clarify the technical novelty of their approach.
Another major issue is the numerous overstatements of the manuscript’s contributions. For example (among many others), Line 21: “this technology has significant implications for pre-hospital environments,” yet the study lacks any human testing, does not involve real-world clinical data, and reports low performance in the real-time model with R² = 0.47, and an accuracy of only 62.4%, well below clinically acceptable thresholds. Similarly, the term “clinically informed” is used in the paper without any definition at all nor supporting evidence. There is no explanation of how clinical knowledge was incorporated into the feature design, nor any analysis of the clinical relevance of the model’s outputs.
Author Response
In this manuscript, the authors aim to non-invasively predict MAP during hemorrhagic shock and resuscitation using PPG in combination with ML models.
We appreciate the reviewer taking the time to provide comments and suggestions for our manuscript. We hope our comments below address the issues, and that it is now suitable for publication.
The primary concern with the paper is that the use of PPG to predict MAP via ML or DL has been extensively explored. The authors themselves cite 5 prior studies that used similar approaches. The main innovation claimed by the authors is the application of this prediction in a hemorrhagic hypotension setting. However, this distinction is relatively minor and not convincingly presented as substantially different from prior work. Specifically, to appropriately motivate the study, the authors should explicitly articulate what and why previous methods fail/would fail to achieve in this specific setting and clarify the technical novelty of their approach.
Yes, the reviewer is correct that the major novelty is trying to predict blood pressure from the PPG waveform during hemorrhage. However, we disagree with the reviewer that this distinction from others’ work focused on normo- and hypertension is minor. There are several physiological changes that occur during hemorrhage including changes to the shape of the PPG waveform as well as your body prioritizing oxygen delivery to central, core organs. As such, there is no reason to think that other PPG to BP prediction models would reasonable work in a hemorrhage setting as the data used to develop these models does not include the waveform changes that occur in these applications.
We have added more description of the physiological differences during hemorrhage and further motivated the completed research study.
Another major issue is the numerous overstatements of the manuscript’s contributions. For example (among many others), Line 21: “this technology has significant implications for pre-hospital environments,” yet the study lacks any human testing, does not involve real-world clinical data, and reports low performance in the real-time model with R² = 0.47, and an accuracy of only 62.4%, well below clinically acceptable thresholds. Similarly, the term “clinically informed” is used in the paper without any definition at all nor supporting evidence. There is no explanation of how clinical knowledge was incorporated into the feature design, nor any analysis of the clinical relevance of the model’s outputs.
We point out limitations to the study in the discussion previously and have now re-worded statements about the contributions of this work to be less matter of fact given the early development stage this work is currently at. However, we do believe the technology with further maturation can improve pre-hospital medicine as stated. We have re-worded the clinically informed language for clarity to point out how the manually extracted features are based on traditional clinical vital sign interpretation whereas the automatic approach does not have this insight. We hope this helps the clarity of these points.
Round 2
Reviewer 2 Report
Comments and Suggestions for Authors
The authors has sufficiently addressed my comments. I don't have further comments and suggest to accept the manuscript in the reivsed form.
Reviewer 3 Report
Comments and Suggestions for Authors
In this new version, the authors added more data and addressed most of the concerns.